# Infrequent Presentations of Chronic *NPM1*-Mutated Myeloid Neoplasms: Clinicopathological Features of Eight Cases from a Single Institution and Review of the Literature

**DOI:** 10.3390/cancers16040705

**Published:** 2024-02-07

**Authors:** Sandra Castaño-Díez, Francesca Guijarro, Mònica López-Guerra, Amanda Isabel Pérez-Valencia, Marta Gómez-Núñez, Dolors Colomer, Marina Díaz-Beyá, Jordi Esteve, María Rozman

**Affiliations:** 1Hematology Department, Hospital Clínic Barcelona, 08036 Barcelona, Spain; scastano@clinic.cat (S.C.-D.); aiperez@clinic.cat (A.I.P.-V.); diazbeya@clinic.cat (M.D.-B.); jesteve@clinic.cat (J.E.); 2Medical School, University of Barcelona, 08036 Barcelona, Spain; 3Institut d’Investigacions Biomèdiques August Pi i Sunyer (IDIBAPS), 08036 Barcelona, Spain; fguijarro@clinic.cat (F.G.); lopez5@clinic.cat (M.L.-G.); dcolomer@clinic.cat (D.C.); 4Hematopathology Section, Servei d’Anatomia Patològica, CDB, Hospital Clínic Barcelona, 08036 Barcelona, Spain; 5Centro de Investigación Biomédica en Red de Cáncer (CIBERONC), 28029 Madrid, Spain; 6Hospital Parc Tauli Sabadell, 08208 Barcelona, Spain; magonu@gmail.com; 7Josep Carreras Leukemia Research Institute, 08916 Badalona, Spain

**Keywords:** *NPM1*, myeloid neoplasm (MN)atypical presentation

## Abstract

**Simple Summary:**

We describe the clinicopathologic features of eight patients with atypical presentation of *NPM1*-mutated myeloid neoplasms (MN) and review the literature. Initially, we extensively describe the rarest case, a patient diagnosed with chronic eosinophilic leukemia (CEL) with less than 1% bone marrow blasts associated with an *NPM1* mutation who progressed to an acute myeloid leukemia (AML). Secondly, we summarize the clinicopathologic features of seven additional cases with infrequent presentation of *NPM1*-mutated MN, five of them corresponding to CMML and the other two to MDS. Thirdly, we review the literature.

**Abstract:**

Non-acute myeloid neoplasms (MNs) with *NPM1* mutations (*NPM1*mut-MNs) pose a diagnostic and therapeutic dilemma, primarily manifesting as chronic myelomonocytic leukemia (CMML) and myelodysplastic syndromes (MDS). The classification and treatment approach for these conditions as acute myeloid leukemia (AML) are debated. We describe eight cases of atypical *NPM1*mut-MNs from our institution and review the literature. We include a rare case of concurrent prostate carcinoma and MN consistent with chronic eosinophilic leukemia, progressing to myeloid sarcoma of the skin. Of the remaining seven cases, five were CMML and two were MDS. *NPM1* mutations occur in 3–5% of CMML and 1–6% of MDS, with an increased likelihood of rapid evolution to AML. Their influence on disease progression varies, and their prognostic significance in non-acute MNs is less established than in AML. Non-acute MNs with *NPM1* mutations may display an aggressive clinical course, emphasizing the need for a comprehensive diagnosis integrating clinical and biological data. Tailoring patient management on an individualized basis, favoring intensive treatment aligned with AML protocols, is crucial, regardless of blast percentage. Research on the impact of *NPM1* mutations in non-acute myeloid neoplasms is ongoing, requiring challenging prospective studies with substantial patient cohorts and extended follow-up periods for validation.

## 1. Introduction

Acute myeloid leukemia (AML) with nucleophosmin (*NPM1*) mutation was fully accepted as a specific AML category in the 2017 World Health Organization (WHO) classification of hematologic neoplasms, with the requirement of a blast cell count ≥ 20% for its diagnosis. Nevertheless, *NPM1* gene mutations rarely occur in other myeloid neoplasms (MNs) with <20% blasts. Current classifications of myeloid neoplasms [1,2] emphasize the importance of genetic work-up, but present overlapping criteria with some differences [1,2,3,4,5]. Notably, the 2022 World Health Organization (WHO) [2] classification has eliminated the blast percentage prerequisite for the diagnosis of *NPM1*-mutated AML though emphasizes the necessity of an integrated diagnosis based on clinicopathologic correlation. In contrast, the 2022 International Consensus Classification (ICC) [1] maintains a 10% blast threshold, as in other AMLs with recurrent genetic alterations.

Nowadays, there is no agreement in the literature on whether non-acute *NPM1*-mutated MNs (*NPM1*mut-MNs) such as chronic myelomonocytic leukemia (CMML) and myelodysplastic syndromes (MDS) with *NPM1*mut should be diagnosed and treated as AML. Recent studies show an aggressive clinical evolution of most cases of non-acute *NPM1*mut-MNs and suggest managing them with intensive AML-type therapeutic protocols whenever possible, including targeted therapy and stem cell transplant [5,6,7,8,9,10,11,12,13,14]. Nevertheless, there are still scarce non-acute *NPM1*mut-MN patients showing prolonged survival under non-intensive management [15].

Furthermore, there are proposals to redefine the blast boundary [14] and create a new MDS/AML category (with 10–30% blasts) [13]. This redefinition aims to provide patients with access to better treatment options, including those in clinical trials, making them eligible for both MDS and AML protocols [13]. These proposals are based on clinical and biological data, suggesting that the distinction between high-risk MDS and AML has no significant impact on clinical outcome in terms of overall survival and event-free survival [13,14]. Nevertheless, these studies also suggest that this new blast boundary (10–30%) is relatively arbitrary too and could be refined based on the mutational profile [13,14]. DiNardo et al. [14] propose considering fitter and younger patients with ≥10% blasts for intensive AML-like treatment, especially when they harbor *NPM1* mutations and/or normal cytogenetics. Conversely, they suggest that older unfit patients with unfavorable genetic risk (complex karyotype, rearranged *MECOM*, *TP53* mutations) should be treated with hypomethylating agents in combination with novel and promising agents such as venetoclax [14]. As the understanding of *NPM1* mutations and their consequences has recently become clearer, various targeting approaches such as menin inhibitors are under investigation [16], particularly for refractory and relapsed AML patients.

The goal of this review is to better understand the significance of *NPM1* mutation in non-acute MN. We herein describe eight patients with infrequent presentation of *NPM1*mut MNs and review the literature. These uncommon cases highlight the requirement for an integrated approach to MN, including clinical, morphological, and molecular data to find the genetic drivers of the disease, which is crucial for a correct diagnosis and risk-adapted treatment decisions.

## 2. An Atypical Presentation of *NPM1*mut MN

A 63-year-old man was diagnosed simultaneously with prostate cancer and a *NPM1*mut myeloid neoplasm in June 2021. He consulted with symptoms of prostatitis and the work-up revealed a prostatic adenocarcinoma together with alterations in the hemogram. Examination was normal. Complete blood count showed macrocytic anemia (MCV 102.6 fL, hemoglobin 106 g/L), with normal platelets (163 × 10^9^/L) and leukocytes (8.24 × 10^9^/L), 39% segmented neutrophils, 1% segmented bands, 19% eosinophils (0.3 × 10^9^/L), 0% basophils, 35% lymphocytes, and 6% monocytes. Marked dysplasia was seen in the eosinophilic elements consisting of nuclear segmentation alterations and wide cytoplasm degranulated spaces, as well as immature granulocytes with eosinophilic granulation (Figure 1a).

These findings led us to make several additional studies, including a bone marrow aspirate and biopsy. The aspirate (Figure 1b,c) was hypercellular and revealed a massive eosinophilic infiltration composed of eosinophilic myelocytes and metamyelocytes with marked dysplasia (sparse granulation or degranulation, abnormal nuclear segmentation, occasional pre-eosinophilic granulation). The red precursors and megakaryocytic elements were slightly decreased without dysplasia and had a normal distribution. Numerous sea blue histiocytes were also observed. Blasts accounted for <1%. Flow cytometry showed that 90% of the bone marrow cells were eosinophils (CD13+, CD33+, CD34−, CD117−, HLA-DR+, CD15+, CD11b+, CD36−, CD66−, NG2−, CD38−, and negative for lymphoid antigens) (Figure 1d). The core biopsy specimen was hypercellular (>90%), with a striking increase in granulocytic cells (positive for CD15 and lysozyme), most of them with eosinophilic differentiation (Figure 1e,f). Megakaryocytes, erythroblasts, mastocytes, and lymphocytes had a normal distribution. Immature neutrophil granulocytic precursors (MPO+) were scarce and located only in the paratrabecular areas. There were <5% of CD34-positive cells. A diffuse loose network of reticulin (MF-1) was observed. Fluorescence in situ hybridization (FISH) excluded both *FIP1L1-PDGFRA* and *PDGFRB* rearrangements. Targeted Next Generation Sequencing (NGS) showed three pathogenic mutations: *TET2* with two variants (c.2671C > T VAF 45.15%, and c.3435delT 44.76%); *NPM1* (c.863_864insTCTG VAF 42.33%); and *SRSF2* (c.284C > T VAF 32.33%). Bone marrow biopsy staining with an anti-*NPM1* antibody showed a cytoplasmic positivity in most of the cells consistent with a delocalization of the protein due to the mutation (courtesy of Dr. Falini) (Figure 1g).

Our diagnosis at that moment was chronic eosinophilic leukemia (CEL) *NPM1*mut vs. acute myeloid leukemia (AML) *NPM1*mut with eosinophilic differentiation. The patient then received radiotherapy over the involved field together with hormone blocking therapy for the prostate neoplasm and was scheduled to start chemotherapy for the MN when he presented subcutaneous nodular lesions on his face. The biopsy of the skin lesions showed a dense infiltration by large cells of blastic appearance that were CD34−, CD15+, CD33+ and MPO−, consistent with a cutaneous sarcoma (Figure 1h). Intensive therapy for the MN was then started and consisted of induction chemotherapy according to the CETLAM-2012 protocol (idarubicin and cytarabine), and one consolidation course with high-dose cytarabine. Therefore, he achieved a morphological complete remission (CR) with a positive molecular measurable residual disease (mut*NPM1*/*ABL1* ratio of 0.0041). Afterwards, he received an allogeneic stem cell transplant (alloHCT) of a matched unrelated donor, with a non-myeloablative conditioning regimen and post-transplant graft-versus-host disease prophylaxis based on cyclophosphamide and tacrolimus. He achieved a complete molecular remission (*NPM1* mut undetectable by qRT-PCR). Thirteen months after alloHCT, he presented with 60% bone marrow blasts with megakaryocytic differentiation (CD36+, CD42a/CD61+, CD42b+, CD41+, CD56+). At that moment, new mutations in *DNMT3A* and *TP53* (two variants) were detected, while original mutations in *TET2*, *SRSF2* were not identified and *NPM1* remained undetectable by qPCR (Figure 1i,j). A chimerism analysis confirmed the recipient origin of this otherwise clonally unrelated second AML. Immunosuppressive therapy was reduced and treatment with Azacytidine and Magrolimab was started, without any response. He died one and a half years after alloHCT.

## 3. Seven Additional Cases of Unusual Presentation of *NPM1*mut MN

Here, we present seven cases of patients diagnosed through a comprehensive evaluation of clinical, morphological, and molecular data, and their treatment was tailored based on risk assessment (Table 1). Five cases were consistent with chronic myelomonocytic leukemia and the other two corresponded to myelodysplastic syndromes. Four of the five CMML cases were included in a multicentric work presented in [17], submitted elsewhere.

Patients with CMML carrying the *NPM1* mutation presented at a relatively young age, with a median of 60 years (range, 55–80). This subgroup exhibited distinctive characteristics, including a median bone marrow blast count of 13% (10–19), leukocytes of 15.7 × 10^9^ (12.24–30.6), a platelet count of 98 × 10^9^ (35–110), and a median hemoglobin concentration of 85 g/L (49–138). Among these five patients, three were eligible for intensive chemotherapy and attained a complete response with a significant reduction in *NPM1* ratio. The same occurred with the patient with an MDS and the case with eosinophilic presentation, both eligible for intensive chemotherapy. All five cases achieved a molecular complete response after allogeneic hematopoietic cell transplantation (*n* = 3) or venetoclax and azacytidine treatment (*n* = 2). In contrast, another patient treated upfront with venetoclax + azacytidine remained with stable disease. Two other patients were older and constrained by comorbidities and could only benefit from cytoreduction with hydroxyurea (*n* = 1) and best supportive care.

## 4. Review

### 4.1. NPM1 in Leukemogenesis and Its Clinical Impact

The nucleophosmin (*NPM1*) gene encodes a multifunctional chaperone protein predominantly situated in the nucleolus. This protein shuttles between the nucleus and the cytoplasm [19,20]. *NPM1* plays pivotal roles in preserving genomic stability, participating in p53-dependent stress responses, facilitating ribosome genesis, and modulating growth suppressive pathways through interactions with Alternate Reading Frame protein (Arf) [21].

Mutations in the *NPM1* gene are found in exon 12 in most cases, consisting of a 4 base pair insertion that causes the loss of a few C-terminal amino acids of the protein [22]. This results in a stronger nuclear export signal, leading to the cytoplasmic localization of the protein, in contrast to the predominantly nucleolus-located wild-type [23]. Aberrant cytoplasmic localization is a shared characteristic of all *NPM1* mutants and is crucial for its role in leukemogenesis [24]. However, the mechanisms governing leukemogenesis remain a subject of debate. It has been proposed that *NPM1* mutations drive leukemia through a combination of the gain and loss of functions in distinct cellular processes at both the nuclear and cytoplasmic levels [25,26,27,28]. In the cytoplasm, mutant *NPM1* may function through two mechanisms: (1) cytoplasmic misplacement of nuclear proteins (ARF, CTCF, FBW7, HEXIM1, MIZ1, and PU.1), although the significance of this process is not yet fully understood; and (2) binding to cytoplasmic proteins caspase-6 and -8, subsequently inhibiting their myeloid differentiation abilities [25,26,27,28]. Recent findings suggest that mutant NPM1 exerts its nuclear-level role by binding to chromatin at the HOXA/B and MEIS1 loci in cooperation with the MLL complex, leading to an increase in the expression of these oncogenic stem cell programs and blocking the normal differentiation of hematopoietic progenitors [29,30]. Aberrant cytoplasmic localization can be easily assessed by immunohistochemistry [11,31,32], and through this method, other rare *NPM1* mutations out of exon 12 and fusion genes involving *NPM1* have been identified, as well as *NPM1* mutations in non-acute MN.

*NPM1* mutations are AML-driving events responsible for around one-third of all AML cases [20], though they have also been identified in non-acute myeloid neoplasms with an initial blast count lower than 20%. It is widely accepted that *NPM1*-mutated AML develops from preexisting clonal hematopoiesis [24], which explains its frequent co-occurrence with *DNMT3A, TET2*, and *IDH*1/2 mutations [10,20].

Even when *NPM1*mut is tightly associated with de novo AML, on rare occasions, *NPM1* mutations can emerge in other myeloid neoplasms as secondary mutations, ultimately contributing to the development of AML, as suggested by clinical studies [4,27] and mouse models [28]. These non-acute MNs have also been regarded as early stage AML [12], due to a mutational profile with fewer myelodysplasia-related gene mutations [5,6], their sensitivity to chemotherapy, the eradication of *NPM1* following intensive treatment and its reappearance at relapse [5,6].

*NPM1* mutations have a strong prognostic significance in the context of de novo AML, defining a favorable risk stratification in the absence of co-occurring *FLT3*-ITD mutations. In these patients, if MRD is cleared after intensive treatment, consolidation with alloHCT is not recommended. However, the prognostic significance of *NPM1* in other myeloid neoplasms remains unknown as well as the best therapy for these patients.

Finally, various *NPM1*-targeted approaches such as menin inhibitors and FLT3 kinase inhibitors are under investigation with promising results [10,33], which warrants the identification of this potential druggable candidate in all patients with a myeloid malignancy [16].

### 4.2. NPM1 Mutation in CMML

CMML is a clonal hematopoietic malignancy characterized by features of both myeloproliferative neoplasms and myelodysplastic syndromes. In this neoplasm the most prevalent mutations include *TET2* (approximately 60%), followed by *SRSF2* (around 50%), *ASXL1* (about 40%), and those related to the oncogenic RAS pathway (approximately 30%) [34,35,36]. Significantly, *ASXL1* and *NRAS* mutations have been identified as indicators of a poorer prognosis [37], while the prognostic implications of *RUNX1* and *SETBP1* remain a subject of ongoing debate [38,39,40,41,42]. While *NPM1* mutation is not a typical hallmark of CMML, it has been infrequently observed in this entity, accounting for 3–5% according to various series [32,43,44,45,46,47,48,49]. The classical type A mutation (c.860_863dup) is found in most cases [48,49]. Studies with paired samples have reported clonal evolution from CMML to AML, revealing the emergence of new mutations and an increase in variant allele frequency (VAF), including *NPM1* [50]. Remarkably, in one of these studies, two out of five evaluable patients revealed no additional cytogenetic or molecular alterations at the moment of AML diagnosis, highlighting the exclusive appearance of the *NPM1* mutation [48]. In the same study, a positive correlation between *DNMT3A, FLT3*-ITD, and *NPM1* in CMML was found [43,48], while a negative association existed with *TET2* and *ASXL1* [48]. Another study [43] reported varying prevalences of *NPM1* and *ASXL1* according to the 2017 WHO classification, with more *ASXL1* mutations in CMML-0 and more *NPM1* mutations in CMML-2.

Previous studies have suggested that *NPM1*mut CMML patients have a higher likelihood of AML transformation, while others have considered them to be in an “early stage” of AML based on shared clinical and molecular characteristics [32,46,48]. In this context, *NPM1*mut CMML patients exhibited lower hemoglobin levels, an increased median leukocyte count (median of 20 × 10^9^/L) [47], elevated bone marrow monocyte and blast percentages, an increased probability of AML evolution, and inferior overall survival compared to *NPM1*wt CMML [44,47,48]. Blast transformation occurred in 62% of cases, with a median time of 5 months after the initial CMML diagnosis. Additionally, AML in NPM1mut progressed patients did not show the favorable prognosis associated with the de novo *NPM1*mut AML [46,48,49,50].

The 2022 ICC classification of MNs [4,8] acknowledges the presence of the *NPM1* mutation in rare CMML cases but notes that these may not be diagnosed with de novo AML, even if the blast count is 10–19% [4]. In contrast, the 2022 WHO classifies these cases as AML with mutated *NPM1* [2]. The management of these CMML cases is challenging, due to the low number of cases there is with a lack of large series [5,9,44,45,48]. Earlier studies suggested a higher probability of AML transformation and poorer overall survival when patients were treated with CMML protocols [47]. In vitro investigations have also provided evidence for the heightened chemosensitivity of leukemic blasts with *NPM1* mutations compared to those with *NPM1* wild-type [22], which has even been demonstrated in *NPM1*mut CMML [5,6,48]. In this sense, Vallapureddy et al. [48] presented a cohort of 373 patients diagnosed with CMML, among whom 8 (2%) had *NPM1* mutation. In this study, five out of eight *NPM1*mut CMML patients progressed to AML at a median of 5 months. Among these, four out of five patients with blast transformation received AML-like induction chemotherapy, and two of them subsequently underwent alloHCT. The median OS after transformation was 3 months, with only one patient who had undergone alloHCT still alive at the last follow-up. Montalban-Bravo et al. [5] demonstrated that individuals with *NPM1*-mutated MDS or MDS/MPN undergoing intensive chemotherapy exhibited elevated overall response rates (100%) and complete response rates (90%). Additionally, they experienced more favorable progression-free survival and overall survival compared to outcomes achieved with hypomethylating agents (HMAs). Nevertheless, the patients treated with chemotherapy were notably younger than those who received HMAs. This age difference could introduce selection bias since these individuals were perceived as more suitable candidates for this treatment, and as a result, better outcomes were anticipated. However, this study was unable to assert that chemotherapy alone represents the optimal strategy due to the short follow-up for the non-transplanted patients. In individuals initially treated with HMAs, consolidation followed by alloHCT demonstrated better survival outcomes, although the transplanted patients were younger. The study by Patel et al. [6] reported unfavorable overall outcomes in their cohort of *NPM1*-mutated myeloid neoplasms (including CMML and MDS), with the majority of patients undergoing initial HMA therapy. Notably, three patients received upfront induction chemotherapy, and none of them developed an AML. These findings led the authors to propose that upfront HMA therapy might have been insufficient for certain patients. These results suggest that suitable patients with *NPM1*mut non-acute MNs with <20% blasts may benefit from AML-type chemotherapy more than the standard CMML approach. Furthermore, alloHCT remains the sole potential curative approach for CMML.

### 4.3. NPM1 Mutation in MDS

The occurrence of *NPM1* mutations in MDS patients is reported to be in the range of 1–6% [7,15,51,52]. Zhang and collaborators [51] were the pioneers in identifying *NPM1* mutations in two MDS patients (5.2%), both of which were type A. Subsequently, Bains et al. [52] reported *NPM1* mutations in seven MDS patients (4.4%), all falling under the refractory anemia with excess blasts category, and with a normal karyotype (*p* < 0.001). They highlighted a significant association between *FLT3* and *NPM1* mutations, with *FLT3* being nearly four times more prevalent in *NPM1*mut MDS patients compared to *NPM1*wt (*p* < 0.001). Although there were no notable differences in terms of progression to AML between *NPM1*mut and *NPM1*wt MDS (*p* = 0.133), the combination of *FLT3*mut and *NPM1*mut adversely impacted progression-free survival (*p* = 0.026). Moreover, the presence of *NPM1*mut in MDS patients has been linked to multilineage dysplasia [12], normal cytogenetics, an excess of blasts [7], CD34 negativity [5], and aggressive clinical evolution with an elevated risk of AML progression [5,52]. Based on these findings, it was suggested that *NPM1* mutation plays a pivotal role in myelodysplasia development [52]. Supporting this claim, the role of *NPM1* heterozygosity in the development of a hematological syndrome resembling human MDS was demonstrated in a mouse model [53,54].

Wu et al. [15] identified 12 high-risk MDS patients (6.2%) carrying the L287fs *NPM1* mutation who received decitabine-based treatment. These patients achieved a higher complete response compared to *NPM1*wt MDS patients (50% vs. 29.1%, *p* = 0.191) and the results were even better for *DNMT3A*wt cases [15]. They also found a good correlation between the mutational burden and the response depth. In patients achieving complete response, *NPM1* mutation was not detected; those with hematological improvement showed a reduction in *NPM1* VAF and non-responsive patients retained an unaltered *NPM1* level [15].

Nonetheless, other studies [5,6] have pointed out that *NPM1*mut MDS patients treated with intensive chemotherapy with or without consolidation with alloHCT (“AML-like” treatment) presented better outcomes, including higher remission rates and better progression-free and overall survival.

### 4.4. NPM1 Mutation in Other Non-Acute Myeloid Neoplasms

The pathogenic involvement of *NPM1*mut in myeloproliferative neoplasms (MPNs) has scarcely been explored so far and remains unclear. Animal models (knock-in mouse model [55,56] and zebrafish [57]) with artificially induced human *NPM1* mutant expression exhibited an increase in hematopoietic cells and the development of myeloproliferation, but this seems not to correlate with patients, as large cohorts of Philadelphia-negative MPNs have failed to find mutations affecting *NPM1*. Nevertheless, cases of *NPM1*mut AML progressing from MPNs have been described, though they are so scarce that no conclusions can be derived about their particular features and outcome [58]. Schittger et al. identified *NPM1* mutations in 6 out of 67 patients with AML (sAML) who had a previous history of MPN. They concluded that the *NPM1* mutation is not solely a crucial factor in the onset of the novo AML but may also play a role in the development of AML subsequent to MPN [59]. Another study proposed that the candidacy of *NPM1* mutation in the development of human MPNs is uncertain and questionable [60]. This uncertainty arises from the absence of *NPM1* hotspot mutation in a relatively larger cohort of MNs, which included 120 classic MPN cases and 9 cases of MDS/MPN. This could be attributed to the possible presence of activating mutations in other non-examined regions of this selected gene A relatively old study [45] with a limited cohort of classic MPNs, including 14 cases of polycythemia vera, 7 cases of essential thrombocythemia, and 9 cases of primary myelofibrosis, was not able to find *NPM1* mutations in these patients. Nevertheless, additional investigations are necessary to elucidate their specific roles in hematopoiesis and their involvement in the pathogenesis of MPN, and to assess their prognostic and therapeutic implications. In the interest of our case with eosinophilic presentation, we have also explored the presence of *NPM1*mut in CEL. CEL is a rare condition characterized by persistent blood eosinophilia (>1.5 × 10^9^/L), without class-defining rearrangements. It was first included as a distinct entity by the 2008 4th edition of the WHO Classification [61]. Information about this condition is limited to case reports [62]. Typically, it presents as an aggressive disorder with unfavorable prognosis, resistance to conventional therapy, and high rates of AML transformation. The most comprehensive case series collected 10 patients [63], and in this study, the reported median overall survival was 22 months. Five patients underwent transformation to acute leukemia, of both myeloid and lymphoid types, with a median time from acute transformation to death of two months [63].

As far as we know, there is only one previous report of CEL with *NPM1*mut [64]. It describes a 70-year-old female who presented with leukocytosis and marked eosinophilia and did not respond to Imatinib. AML transformation occurred five months after initial diagnosis, with the identification of an *NPM1* type A mutation (c.860_863dup) together with mutations affecting *TET2* and *FLT3*-TKD. The retrospective analysis of the *NPM1* mutation showed that it was already present at the initial CEL diagnosis, with a high variant allele frequency. The patient did not respond to decitabine and passed away because of a subdural hematoma.

## 5. Conclusions and Future

Non-acute *NPM1*-mutated MNs (*NPM1*mut-MNs) represent both a diagnostic and therapeutic challenge. Most of them correspond to chronic myelomonocytic leukemia (CMML) and myelodysplastic syndromes (MDS), and it is still controversial as to whether they should be diagnosed and/or treated as AML.

Moreover, exploring the role of *NPM1* mutation in non-acute myeloid neoplasms is an evolving area of research, with the possibility of new insights emerging over time. The impact of this mutation on disease progression and patient outcomes varies depending on the specific type of myeloid disorder. Moreover, the prognostic significance of *NPM1* mutations in non-acute MNs is not as firmly established as it is in AML. The notably aggressive clinical course observed in non-acute MNs with *NPM1* mutation, leading to a frequent and rapid evolution to AML, highlights the necessity for a comprehensive diagnosis that incorporates both clinical and biological data. This reinforces the clinical rationale to customize the management of these patients on an individualized basis, opting for intensive treatment more aligned with AML protocols, regardless of blast percentage. Therefore, it is essential and challenging to validate these hypotheses and findings through prospective studies that include a substantial number of patients with a long follow-up period.

## Figures and Tables

**Figure 1 cancers-16-00705-f001:**
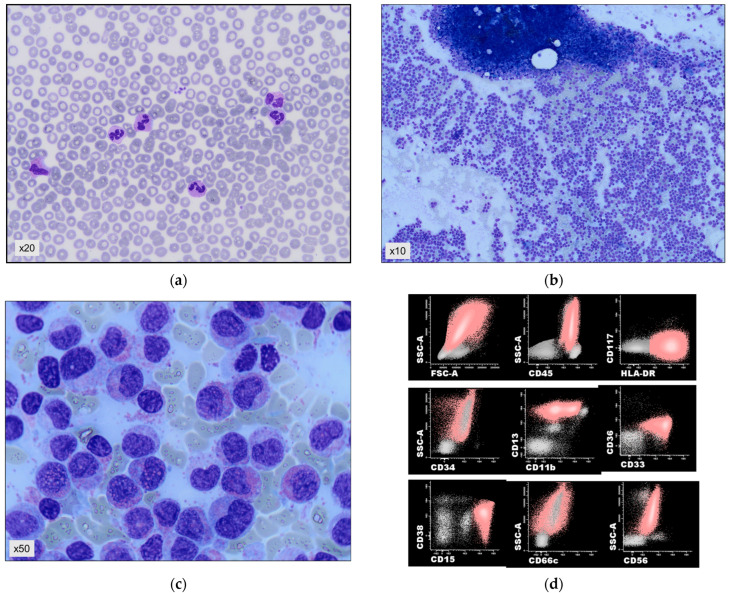
Representative images of the main results of the hematopathologic work-up: (**a**) peripheral blood smear showing atypical eosinophils; (**b**,**c**) bone marrow aspirate markedly hypercellular with massive infiltration by atypical eosinophilic semi-mature granulocytes; (**d**) flow cytometry plots showing positivity for CD15 and HLA-DR and negativity for CD117: the pathological population is represented in pink; (**e**) bone marrow biopsy, the atypical infiltrate shows intense positivity for CD15 and negativity for myeloperoxidase and CD34; (**f**) skin biopsy; (**g**) bone marrow biopsy, diffuse cytoplasmic positivity of the cells for the *NPM1* antibody (courtesy of Dr. Falini); (**h**) skin biopsy with monomorphic infiltration by blasts; (**i**,**j**) fishplot and variant allele frequency (VAF) representation of the mutational pattern at CEL diagnosis and AML progression. Symbols are used to differentiate variants of mutations (* and ^). The magnification factor is displayed in each figure.

**Table 1 cancers-16-00705-t001:** Clinicopathological characteristics of the 8 patients with non-acute *NPM1*-mutated myeloid neoplasms included in the study.

Patient ID (Age, Yrs/Sex)	ICC and WHO 2022 Classification	WBC (×10^9^/L)	PLT (×10^9^/L)	Hb (g/L)	BMB (%)	Cytology *	Karyotype	Mutations Detected	Treatment for mut*NPM1*-MN and Response	Status at Last Follow-Up (Survival in Months)
*NPM1*mut-CMML										
55/F	CMML-2 ^	14.95	98	98	10–15 (BM biopsy)	Granulocytic dysplasia	46, XX [18]	*NPM1*, *CEBPA*, *DNMT3A*, *FLT3-*TKD, and *IDH1*	(1) ESA; (2) Idarubicin + Cytarabine + Midostaurin followed by alloHCT (mCR); (3) Sorafenib (mCR)	Dead in remission (28)
56/M	CMML-2 ^	12.24	70	85	19	Three-lineage dysplasia	46, XY, t(14;15)(q32;q22) [18]	*NPM1*, *FLT3*-TKD, *FLT3*-ITD, and *TET2*	(1) Azacytidine x1 (SD); (2)Idarubicin + Cytarabine + Midostaurin followed by HiDAC x2 (CR); (3)Venetoclax + Azacytidine x1 followed by sequential alloHCT (mCR)	Dead with disease (29)
73/F	CMML-2 ^	30.60	110	49	16	Granulocytic dysplasia	46, XX [18]	*NPM1*, *DNMT3* (2 variants), and *FLT3*	Hydroxyurea (PD)	Death with disease (6)
60/M	CMML-2 ^	27.93	35	58	13	Granulocytic dysplasia	46, XY [18]	*NPM1* and *DNMT3A* (2 variants)	(1) Idarubicin + Cytarabine (mCR); (2) Venetoclax + Azacytidine x12 (mCR)	Alive without disease (39)
80/M	CMML-1 + BPDCN	15.71	101	138	11	Three-lineage dysplasia	45, X, -Y	*CALR*, *MPL*, *NRAS*, *PHF6*, and *TET2* (2 variants) (bone marrow)*/NPM1*, *CALR*, *NRAS*, and *TET2* (2 variants) (skin)	Venetoclax + Azacytidine (SD)	Dead with disease (12)
*NPM1*mut-MDS										
71/F	MDS/AML, MDS-IB	2.4	146	117	15	Granulocytic dysplasia	46, XX	*NPM1* and *TET2*	(1) Idarubicin + Cytarabine followed by HiDACx1 (CR); (2)Venetoclax + Azacytidine x12 (mCR)	Alive in remission (17)
84/F	MDS NOS, MDS-LB	8.47	81	132	4	Megakaryocytic and granulocytic dysplasia	46, XX	*NPM1*, *EZH2*, *TET2*, and *STAG2*	Transfusion support	Dead with disease (16)
Others *NPM1*mut-MN										
63/M	Chronic eosinophilic leukemia vs. AML	3.25	70	111	0	Granulocytic dysplasia	NA	*NPM1*, *SRSF2*, and *TET2*	(1) Idarubicin + Cytarabine followed by HiDAC x1 (CR) and consolidated with alloHCT (mCR); (2) Azacytidine + magrolimab (PD)	Dead with disease (24)

* Type of lineage dysplasia. yrs, years; M, male; F, female; WBC, white blood cells; PLT, platelets; Hb, hemoglobin; BMB, bone marrow blast; NA, not available; ESA, Erythropoiesis-Stimulating Agents; HiDAC, high-dose cytarabine; alloHCT, allogenic hematopoietic cell transplantation; CR, complete remission; mCR, molecular complete remission; PD, progressive disease; SD, stable disease; NOS, not otherwise specified; LB, low blasts; BPDCN, Blastic Plasmacytoid Dendritic Cell Neoplasm. ^ These CMML cases were included in [17].

## Data Availability

Due to privacy and ethical concerns, the data that support the findings of this study are available on request from the corresponding author.

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
