# Peer review of "Infrequent Presentations of Chronic NPM1-Mutated Myeloid Neoplasms: Clinicopathological Features of Eight Cases from a Single Institution and Review of the Literature"

_cancers, 2024, doi:10.3390/cancers16040705_

Round 1

Reviewer 1 Report

Comments and Suggestions for Authors

It's an interesting well-written as well as well referenced paper. The manuscript would be very useful for the reader's journal. I have no criticisms.  

Author Response

It's an interesting well-written as well as well referenced paper. The manuscript would be very useful for the reader's journal. I have no criticisms. 

Our Response: Dear Reviewer #1, thank you for your encouraging observation.

Reviewer 2 Report

Comments and Suggestions for Authors

good review of a rare entity: NPM1 myeloid neoplasms before AML

may be the word chronic should be mentioned

the authors should takle the issue of these rare cases

minor modifications during the typing:

1)  table 1 third case of CMLL "dead withdisease" as the rest instead of "death"

page9 lane 310 "between" instead of "bwtween"

page10

lane 335 space beween with and a

lane 340 a space should be removed between "therapeutic implications"

Author Response

Good review of a rare entity: NPM1 myeloid neoplasms before AML. May be the word chronic should be mentioned.

Our Response: Dear Reviewer #2, thank you for your positive feedback. We have incorporated your suggestion and added the word "chronic" to the manuscript title, as follows:

(Manuscript, page 1, line 2, Title)Infrequent Presentations of Chronic NPM1-Mutated Myeloid Neoplasms: Clinicopathological Features of 8 Cases from a Single Institution and Review of the Literature

The authors should takle the issue of these rare cases minor modifications during the typing:

1) table 1 third case of CMLL "dead withdisease" as the rest instead of "death"

2) page 9, line 310 "between" instead of "bwtween"

3) page10, line 335 space between with and a

4) page 10, line 340, a space should be removed between "therapeutic implications"

Our Response: thank you for your constructive suggestions. We have implemented the recommended changes from this reviewer in the manuscript.

Please let us know if you require any further information at this time.

Looking forward to hearing from you.

Reviewer 3 Report

Comments and Suggestions for Authors

The manuscript by Castano-Diez et al. discusses the diagnosis and treatment of non-acute NPM1-mutated myeloid neoplasms describing their own clinical experience and reviewing the literature on the published data. The manuscript may be of interest to both clinicians and researchers. In general, the article makes a positive impression, but there are several comments.

Section 2. An atypical presentation of NPM1mut MN

* It is necessary to add magnification factor for photos with cells in Figure 1 (a, b, c, e-h). Also, it would be good to make the same height for the Fig.1E and the Fig.1F, and to align Figures 1C, 1E, 1G to the left margin.

* According to the described data about the case of chronic eosinophilic leukemia with NPM1 mutation after alloHCT new mutations in DNMT3A and TP53 (two variants) were detected, while original mutations in TET2, SRSF2 were not identified. Usually the DNMT3A, TET2 mutations appear at the early stages of the clonal evolution. How do the authors explain the data on such mutational changes?

For FACS Figure 1D only few markers were shown but a very rich signature for eosinophils was discussed in the text of the manuscript.

Section 3. Seven additional cases of unusual presentation of NPM1mut MN

* Summarizing data on non-acute NPM1-mutated MNs in the form of a table is good idea. But the Table 1 is presented in an awkward way. It would be better to modify Table 1, e.g. to add lines for columns or to present the table in landscape mode with larger spacing between words at the borders of the columns.

* I wouldn't make any predictions about trends in age, number of blasts, etc. with such a small group of patients (5 for CMML). I think if the authors add and discuss unique and noteworthy features of these clinical cases in the text, it will increase the value of this manuscript.

*Did patients with NPM1-mutated CMML and MDS have changes in NPM1 mutation status at onset, remission, relapse?

Section 4.1. NPM1 in leukemogenesis and its clinical impact

 *References for the following sentences should be added to the literature review:

1.       Mutations in the NPM1 gene are most commonly found in exon 12, consisting of a 4 base pair insertion that results in the loss of some C-terminal amino acids of the protein.

2.       Aberrant cytoplasmic localization is a common feature of all NPM1 mutants and is critical for its role in leukemogenesis.

3.       It has been proposed that NPM1 mutations drive leukemogenesis through a combination of gain and loss of function in different cellular processes at both the nuclear and cytoplasmic levels.

 Section 4.2. NPM1 mutation in CMML

“Notably, ASXL1, NRAS, RUNX1, and SETBP1 mutations have been identified as indicators of a worse prognosis [36].”

The prognostic significance of most mutations in CMML is controversial. I recommend not to write about RUNX1 and SETB1 as markers of a worse prognosis. For the RUNX1 mutations there are controversial data about OS [DOI: 10.1016/j.ebiom.2018.04.018, DOI: 10.1038/leu.2009.48]  although trend towards increased progression to AML [DOI: 10.1038/leu.2009.48].

For SETBP1 there are controversial data about OS and its impact on progression to AML [PMC6886439, PMC3806243, DOI: 10.1038/leu.2014.125, DOI: 10.1038/bcj.2014.90, DOI: 10.1371/journal.pone.0171608].

Section 4.3. NPM1 mutation in MDS

Misprints in the following words should be corrected:

Line 310 - between, depth;

Line 335 – “witha”;

Line 337 – “findNPM1”;

line 347 – “colleced”.

Author Response

Section 2. An atypical presentation of NPM1mut MN

*It is necessary to add magnification factor for photos with cells in Figure 1 (a, b, c, e-h). Also, it would be good to make the same height for the Fig.1E and the Fig.1F, and to align Figures 1C, 1E, 1G to the left margin.

Our Response: Dear Reviewer #3, thanks for your positive comments and for pointing out the importance of including a magnification factor for the photos, ensuring consistent figure height, and aligning them with the margins. We deeply value your suggestions, and we have incorporated them into the manuscript.

*According to the described data about the case of chronic eosinophilic leukemia with NPM1 mutation after alloHCT new mutations in DNMT3A and TP53 (two variants) were detected, while original mutations in TET2, SRSF2 were not identified. Usually the DNMT3A, TET2 mutations appear at the early stages of the clonal evolution. How do the authors explain the data on such mutational changes?

Our Response: Indeed, we were greatly surprised by the mutations found at relapse, so that we even considered the possibility we were facing a donor-related acute leukemia. A chimerism analysis, also performed on tumor-purified CD34+ cells, confirmed the recipient origin of the relapse. After alloHCT, the clones carrying mutations in TET2 and SRSF2, may have been effectively eliminated. The clones with DNMT3A and TP53 could have been selected and expanded under therapy pressure from initially undetected low-frequency clones, as our NGS diagnostic panel can only detect mutations with variant allele frequency over 1%.

The occurrence of AML with wild-type NPM1 (wtNPM1-AML) following complete remission (CR) of mutNPM1-AML has been observed in approximately 10% of patients [1-3]. However, the progression from a non-AML wild-type NPM1-myeloid neoplasm (MN) to wtNPM1-AML has not been documented. As far as our knowledge extends, the incidence of this phenomenon has yet to be explored in a large and uniformly treated cohort of patients with long-term follow-up.

Sequential genomic studies, not confined to the NPM1 founder mutation but employing next-generation sequencing (NGS), have the potential to unveil the emergence of wtNPM1-MN and shed light on its clonal origin [3,4].

We emphasize the importance of notifying physicians about this potential evolution to ensure the proper monitoring of patients exhibiting this type of condition.

*For FACS Figure 1D only few markers were shown but a very rich signature for eosinophils was discussed in the text of the manuscript.

Our Response: In accordance with the feedback from Reviewer #3, we provide a comprehensive list of eosinophil markers in Figure 1D.

Section 3. Seven additional cases of unusual presentation of NPM1mut MN

*Summarizing data on non-acute NPM1-mutated MNs in the form of a table is good idea. But the Table 1 is presented in an awkward way. It would be better to modify Table 1, e.g. to add lines for columns or to present the table in landscape mode with larger spacing between words at the borders of the columns.

Our Response: Following the suggestion put forth by Reviewer #3, we have modified the table to landscape mode, improving its visibility.

*I wouldn't make any predictions about trends in age, number of blasts, etc. with such a small group of patients (5 for CMML). I think if the authors add and discuss unique and noteworthy features of these clinical cases in the text, it will increase the value of this manuscript.

Our Response: In accordance with the suggestions from reviewer #3, I have removed the predictions due to the small number of patients and emphasized the clinical significance of these cases.

(Manuscript, pages 8, lines 164-205)

Patients with CMML carrying the NPM1 mutation presented at a relatively young age, with a median of 60 years (range, 55-80). This subgroup exhibited distinctive characteristics, including a median bone marrow blast count of 13% (10-19), leukocytes of 15.7 x 10^9 (12.24-30.6), a platelet count of 98 x 10^9 (35-110), and a median hemoglobin concentration of 85 g/L (49-138). Among these five patients, three were eligible for intensive chemotherapy and attained a complete response with a significant reduction in NPM1 ratio. The same occured with an MDS and the case with eosinophilic presentation, both eligible for intensive chemotherapy. All five cases achieved a molecular complete response after allogeneic hematopoietic cell transplantation (n=3) or venetoclax and azacytidine treatment (n=2). In contrast, another patient treated upfront with venetoclax + azacytidine remained with stable disease. Two other patients were older and constrained by comorbidities and could only benefit cytoreduction with hydroxiurea (n=1) and best supportive care.

*Did patients with NPM1-mutated CMML and MDS have changes in NPM1 mutation status at onset, remission, relapse?

Our Response:

Certainly, patients diagnosed with Chronic Myelomonocytic Leukemia (CMML) and Myelodysplastic Syndromes (MDS) exhibiting the NPM1 mutation underwent dynamic changes in the mutation's status throughout the course of their diseases. We have included this information in the aforementioned paragraph, but outlined below you can find detailed information about each patient:

Patient One (CMML): Induction chemotherapy resulted in complete remission (CR) with a mutNPM1/ABL1 ratio of 0.023. Allogeneic Hematopoietic Cell Transplantation (alloHCT) from a matched unrelated donor was performed, achiving molecular CR (mutNPM1/ABL1 ratio of 0) one month post-transplant. Three months later, molecular relapse occurred (NPM1 with VAF 2.3%)  with an emergent FLT3-ITD mutation. Sorafenib treatment induced a complete molecular remission lasting a year. Death occurred 1.5 years after alloHCT, attributed to septic shock, with CR persisting at autopsy.

Patient Two (CMML): Initially treated with one cycle azacytidine. Induction chemotherapy led to complete remission with detectable minimal residual disease (mutNPM1/ABL ratio 2.48). This patient underwent a sequential alloHCT, achieving a molecular complete remission. Unfortunately, he relapsed one and a half years later.

Patient Three (CMML): He could only receive hydroxiurea with progressive disease.  NPM1 status was not further assessed.

Patient Four (CMML): Intensive chemotherapy achieved complete molecular remission, but an early molecular relapse occurred four months later. Venetoclax + azacytidine salvage treatment led to a sustained complete molecular remission that persists until now.

Patient Five (CMML): Received venetoclax and azacytidine, but only achieved stable disease and died one year after diagnosis.

First MDS Patient: Intensive chemotherapy resulted in complete remission but with persistent minimal residual disease (NPM1/ABL ratio 0.1). Venetoclax and azacytidine salvage treatment induced a complete molecular remission thereafter.

Second MDS Patient: This patient was initially diagnosed of erythroblastopenia and treated with azathioprine. Afterwards she was diagnosed with MDS with low blast with the finding of the NPM1 mutation. Due to age and comorbidities, no intensive treatment was administered, and re-evaluation of NPM1 status was not conducted. We have changed the treatment described in the table, as azathioprine was previous to MDS diagnosis.

Finally, the clinical and mutational evolution of NPM1 in the patient diagnosed with chronic eosinophilic leukemia versus Acute Myeloid Leukemia (AML) is elaborated in Section 2, titled "An Atypical Presentation of NPM1mut MN."

Section 4.1. NPM1 in leukemogenesis and its clinical impact

*References for the following sentences should be added to the literature review:

  1. Mutations in the NPM1 gene are most commonly found in exon 12, consisting of a 4 base pair insertion that results in the loss of some C-terminal amino acids of the protein.

  1. Aberrant cytoplasmic localization is a common feature of all NPM1 mutants and is critical for its role in leukemogenesis.

  1. It has been proposed that NPM1 mutations drive leukemogenesis through a combination of gain and loss of function in different cellular processes at both the nuclear and cytoplasmic levels.

Our Response: Following the recommendations of Reviewer #3, we have incorporated the references for the preceding statements and subsequently made corresponding modifications to the manuscript. The specific modifications have been highlighted in blue for clarity:

(Manuscript, page 8, lines 215-226)

  1. Mutations in the NPM1 gene are found in exon 12 in most cases, consisting of a 4 base pair insertion that causes the loss of a few C-terminal amino acids of the protein [21].
  2. Aberrant cytoplasmic localization is a shared characteristic of all NPM1 mutants and is crucial for its role in leukemogenesis [23].
  3. It has been proposed that NPM1 mutations drive leukemia through a combination of gain and loss of functions in distinct cellular processes at both the nuclear and cytoplasmic levels [24-27].

Section 4.2. NPM1 mutation in CMML

“Notably, ASXL1, NRAS, RUNX1, and SETBP1 mutations have been identified as indicators of a worse prognosis [36].”

The prognostic significance of most mutations in CMML is controversial. I recommend not to write about RUNX1 and SETB1 as markers of a worse prognosis. For the RUNX1 mutations there are controversial data about OS [DOI: 10.1016/j.ebiom.2018.04.018, DOI: 10.1038/leu.2009.48]  although trend towards increased progression to AML [DOI: 10.1038/leu.2009.48].

For SETBP1 there are controversial data about OS and its impact on progression to AML [PMC6886439, PMC3806243, DOI: 10.1038/leu.2014.125, DOI: 10.1038/bcj.2014.90, DOI: 10.1371/journal.pone.0171608].

Our Response:

Certainly, several pivotal studies have contributed to establishing risk scores in the field of CMML, including the CPSS (Such et al., Blood, 2013, Apr 11;121(15):3005-15. doi: 10.1182/blood-2012-08-452938), CPSS-Mol (Elena et al., Blood, 2016, Sep 8;128(10):1408-17. doi: 10.1182/blood-2016-05-714030), and the Mayo prognostic model (Patnaik et al., Leukemia, 2013, Jul;27(7):1504-10. doi: 10.1038/leu.2013.88.). These studies conducted multivariate analyses to assess the independence of variables in relation to other significant factors.

In response to concerns raised by Reviewer #3, particularly regarding the ongoing debate surrounding the prognostic impact of RUNX1 and SETBP1 mutations in CMML patients, we have revised the relevant sentence in the manuscript as follows:

(Manuscript, page 9, lines 260-263)

Significantly, ASXL1 and NRAS mutations have been identified as indicators of a poorer prognosis [37], while the prognostic implications of RUNX1 and SETBP1 remain a subject of ongoing debate [38-42]”

Section 4.3. NPM1 mutation in MDS

Misprints in the following words should be corrected:

Line 310 - between, depth;

Line 335 – “witha”;

Line 337 – “findNPM1”;

line 347 – “colleced”

Our Response: In accordance with the feedback from Reviewer #3, we have identified and rectified typographical errors in the designated sections of the manuscript.

Please let us know if you require any further information at this time.

Looking forward to hearing from you.

Reviewer 4 Report

Comments and Suggestions for Authors

The manuscript presented contain significant results, however in my opinion it is not well organized.

The manuscript present 8 cases with rare presentation from the genetically point of view. However, it is well know all of this (see for example PMID28707414).

Moreover, the authors decided to present just 8 cases and not all their results presented on an abstract (Volume 142, Supplement 1, 2 November 2023, Page 3231). It will be interesting to present all of them and not to split the results several times.

The Review part do not have significant contribution to the field!

I suggest to reject this paper and I highly recommend  to present the whole genomic data of all your patients. 

Author Response

Our Response:

We are sorry you could not find enough interest in our work. Indeed, the presence of NPM1 mutation in myeloid neoplasms other than AML has been long described. However, we believe this is not a closed subject that still deserves consideration as shown by the inconsistence regarding the blast count required for AML with NPM1 mutation diagnosis between WHO and ICC classifications. Recent publications written by distinguished experts in the field show how this debate is still open (Falini B, et al. Am J Hematol. 2023 Jul;98(7):E187-E189. doi: 10.1002/ajh.26946. PMID: 37119006).

In reference to the mentioned abstract, its purpose markedly differs from our work. Our manuscript wants to emphasise the diagnostic problems that non-acute NPM1 mutated cases may present whereas the abstract represent a multicentric work and pretends to collect CMML cases with mutations associated to AML. Moreover, the abstract includes cases collected from a comprehensive investigation conducted across various Spanish medical centers that take part of Grupo Español de Síndromes Mielodisplásicos (GESMD). This work was focused specifically on cases of Chronic Myelomonocytic Leukemia (CMML) characterized by mutations commonly associated with Acute Myeloid Leukemia (AML). In contrast, our study meticulously examines non-acute myeloid neoplasms, and draw data exclusively from a single medical center. Of particular significance is a conspicuous case with eosinophilic presentation featuring a mutation in NPM1, an occurrence considered atypical within this context. We think we give an important message about how other uncommon presentations of myeloid neoplasms can also harbour NPM1 mutation. Our objective is to alert diagnosticians to be aware that these rare cases exist, urging them to remain vigilant to avoid errors. The reference to the abstract has been incorporated into our manuscript.

The review aimed to gather all relevant publications addressing this subject. While most works are focused on MDS and CMML, we also included a section dedicated to other myeloid neoplasms with NPM1 mutation.

While existing literature extensively covers mutations associated with CMML and MDS, as evidenced by cited references in our manuscript, our comprehensive review primarily explores the clinical dimensions of these cases, with a pronounced focus on the foremost exceptional case, all derived from a singular center. This nuanced approach not only contributes to the broader understanding of non-acute myeloid neoplasms but also accentuates the rarity and clinical implications of the highlighted cases, thereby enriching the scientific discourse surrounding hematological malignancies with NPM1 mutations.

Please let us know if you require any further information at this time.

Looking forward to hearing from you.